# Sustainability of translator training in higher education

**Minghai Zhu** *

Guangdong Open University (Guangdong Polytechnic Institute), School of Applied Foreign Languages, Guangzhou, China

* 308959821@qq.com

**Data Availability Statement:** All relevant data are within the paper and its Supporting Information files.

**Funding:** This research was funded by Guangdong Open University (Guangdong Polytechnic Institute), grant number ZD1902 (URL:http://www.gdrtvu.

## Abstract

The United Nations has set a Sustainable Development Goal in education to be met hopefully by 2030. One of the target areas is to substantially increase the number of youth and adults possessing training and proficiency relevant to the technical and vocational skills needed for employment, well-paying jobs, and decent entrepreneurships. Enrolled students need to be equipped with core competencies suitable for the fields in which they are specializing, including the field of translation. For student translators, "transcreation" is a core competency they are expected to acquire and practice. With its increasing application in all sectors of life, the use of artificial intelligence or machine translation is on its way to becoming mainstream in the translation industry, eliminating bread-winning opportunities from translators, leaving them in the stream of life to sink or swim. That is why trainers of translators and practitioners alike insist that it is time to consider transcreation so that student translators can better embrace future challenges and boost their employability. A one-shot case study was adopted in this research. After a one-semester trial of teaching and practicing transcreation, an online questionnaire survey was administered to gain the overall perceptions of transcreation from the students. Findings show that the students have raised their awareness of transcreation as a novel approach to translation and most of them feel confident about their employability in the translation job market. Implications for translation syllabus design and translator training are also illustrated.

## Introduction

In 2015, the United Nations established 17 Sustainable Development Goals as a part of Agenda 2030, which is an urgent call for action by all member countries involved [1]. One of the target areas in education is to substantially increase the number of youth and adults possessing relevant training and proficiency, including technical and vocational skills, to help them find employment with reasonable incomes or all them to develop beneficial entrepreneurships [2]. Knowing how to equip young people and adults with those skills is a major challenge for educators and trainers around the globe. Students need to be equipped with core competencies suitable for the fields in which they are specializing including the field of translation. "Transcreation," which means translating creatively, is a core competency student translator are expected to acquire and practice. Translator educators and trainers need to keep informed of

edu.cn/) for Minghai Zhu. The funders played no role in the design, data collection and analysis of this research, decision to publish, or preparation of the manuscript.

**Competing interests:** The authors have declared that no competing interests exist.

the latest competencies required of translators by the translation industry so as to give student translators adequate training in those competencies, which in turn will empower them in the job market [3].

First, let us consider a basic question, what is transcreation? A simple definition is available at Google [4]:

"Transcreation is a fusion of the words "translation" and "creation". It describes copywriting content in a source text that needs to be made coherent, relevant, etc. in a new language. Sometimes transcreation is also called "creative translation." Namely, because the content isn't translated word for word."

Díaz-Millón and Olvera-Lobo [5], based on previous academic conceptualizations of transcreation, present a broad definition as follows, with the goal of including all the major features identified in the relevant literature [6–8]:

"Transcreation is a type of translation characterized by the intra-/interlingual adaptation or re-interpretation of a message intended to suit a target audience, while conveying the same message, style, tone, images and emotions from the source language to the target language, paying special attention to the cultural characteristics of the target audience. This re-interpretation of the message may imply adaptations that move away from the original text to a greater or lesser extent to fit the original purpose, transmit the original message and overcome cultural barriers. For such reasons, it is present in persuasive and communicative contexts."

That definition is comprehensive but far too long although many other definitions are available [6,9]. It is no simple task to come up with a concise yet universally accepted definition because the theme would require a full-length paper to explore. For the convenience of the present discussion, the present author defines the term in a simplified way:

Transcreation, a form of rewriting or copywriting in cross-cultural promotions like advertising, marketing and literary translation or for other international communicative purposes, deviates from the source text to a lesser or greater extent, or even completely, so as to better serve the target audience.

However, a key concept involved in this definition needs to be clarified before the entire discussion can move on: What is culture? After all, the differences between the source culture and the target culture embodied by the source audience and the target audience, respectively, are what triggers the need for transcreation.

The heart of a culture involves language, religion, values, traditions and customs [10]. Culture has proved to be quite difficult to define because more than 100 definitions of culture exist [11]. Abi-Hashem [12] offered this definition:

"Culture shapes the life of the community and in return is shaped by the community itself. It is, at the same time, the cause and the outcome, the source and the product. Cultures have an abstract and a concrete element to them. They are, at once, tangible and symbolic, moral and temporal. They represent connectivity with the past and continuity into the future".

John W. Berry et al. [13] simply define culture as "the shared way of life of a group of people" (p. 2). Kenneth D. Keith [14] stresses that it is "important to note what culture is not.

Perhaps most importantly, culture is not synonymous with nationality or race." According to Fiske [15], culture consists of ideas, symbols, values, practices, and competencies. Of all competencies, cultural competency is extremely important in cross-cultural or inter-cultural communication. D. W. Sue et al. [16] formulated the most widely recognized conceptual framework of cultural competency covering three major areas: cultural awareness and beliefs, cultural knowledge (including knowledge of target worldviews) and cultural skills. Without a sharp awareness of their own cultural roots, cross-cultural professionals risk imposing their assumptions, concepts, practices and values on others from different cultural backgrounds [17]. Therefore, the practice of cultural self-awareness is still desirable [18]. Worldview is one's outlook about life and "the process of discovering and understanding a different worldview can actually be a cross-cultural experience. Such encounter requires the skills of empathy and cultural sensitivity" [19]. Stanley Sue [20] went on to make proposals on how to translate cultural competency from a philosophical definition into practice. Abi-Hashem [21] notes that "cultural competency has been highly encouraged and emphasised in all areas of professional services and interactions." The training of such a competency is required of professionals who work in cross-cultural settings [22,23]. The same is applicable to student translators who are sure to work cross-culturally.

According to Abi-Hashem [21], cultures are "better felt than defined and better experienced than explained" (p. 26). Most student translators do not have the privilege to feel and experience foreign cultures in person. Instead, they can learn about other cultures by reading and becoming informed by cultural anthropology including psychological anthropology whose insights are very useful to professionals engaged in cross-cultural activities [24]. Anthropology is the study of humanity while "culture is the glue that holds humans together. Therefore, cultural anthropology is the combination of these constructs that allow one to gain a better understanding of various groups" [25]. As a branch of cultural anthropology, psychological anthropology focuses on the effects culture has on the individual and vice versa [26]. Large amounts of relevant information are available online. Students can also watch relevant movies or videos online. Teachers or trainers of translators can give student translators many recommendations in this respect.

Undeniably, machine translation has benefited translators enormously and benefited the translation industry as well. According to Yorick Wilks [27], "MT [machine translation] systems, for example at the Federal Translation Division in Dayton, Ohio and the European Commission in Luxembourg, produce fully automatic translations on a large scale that many people use with apparent benefit" (p.1). Amazon Web Services (AWS) provides a working definition of machine translation [28]:

> "Machine translation is the process of using artificial intelligence to automatically translate text from one language to another without human involvement. Modern machine translation goes beyond simple word-to-word translation to communicate the full meaning of the original language text in the target language. It analyzes all text elements and recognizes how the words influence one another".

Machine translation has improved the work efficiency of translator like never before. For the translation industry, labor costs have been greatly reduced. Studies show that machine translation has increased the productivity of human translators by more than 50% [29]. A common mode of human–machine interaction during the MT process works like this: First, the source text is input; then the MT software produces an output as a draft that can then be corrected by a human translator. This process is often referred to as post-editing (MTPE, or simply PE) [30]. Researchers are optimistic that smart or intelligent machines will replace and

enhance human abilities in many areas [31]. In fact, MT is currently widely used in many fields due to its low cost, high efficiency and ever-improvement of translation quality [32]. "In China, typical human translation costs from 0.1 to 0.5 CNY per character, depending on the translator's proficiency, whereas MT systems cost about 0.00005 CNY per character" [32]. Because of this, "there are growing fears of machine translation taking over translator jobs" [33]. Obviously, machine translation poses huge challenges which seriously threaten the survival of employment for translators and the sustainability, not merely of university programs in translation, but of the translation industry as a whole. Some insist that translators in the new era should develop new skills and adapt to the changing needs of the market, thus advocating versatility in translator training [34]. But just because there are mobile apps or online engines that engage in translation from local-national into global or international languages, this does not necessarily eliminate the need for human translators. The relationship between human translators and artificial intelligence (AI) is not a question of either. . .or but a matter of both; that is, human translators and AI together will meet future needs. Smart or intelligent machines should be treated as colleagues rather than rivals and no need exists to "race against a machine" [35]. Now that AI has taken over a large share of repetitive or routine work, human translators need to focus on their own creativity. I. Mihalache [36] insists that translators need to improve their education and skills by constant training in softer skills such as creativity (p. 33). According to A. Sakamoto [37], "[t]he fact that machines are increasingly outpacing humans—not always in terms of quality, but certainly in speed and volume—is one of the reasons. . .that translation is being devalued" (p. 240). Similarly, according to Katan [38], that fact that AI or machine translation engines such as Google Translate are increasingly marginalizing professional translators means that fewer job opportunities will exist in the field of translation unless the translator are equipped with value-added competences and are able to offer value-added services.

The problem that professional and student translators have encountered is now clear; while working in parallel or rather in collaboration with AI or machine translation, many of them feel devalued and do not know how to react.

In view of this problem, the present study reviews the literature from three perspectives: (1) practitioners and translation agencies (or language service providers (LSPs) as they are now popularly called), (2) standardization organizations such as the International Organization for Standardization (ISO), and (3) researchers, trainers and institutions of higher education.

Benetello [39] believes that conventional translation approaches very often do not work in the age of AI (including machine translation) and thus strongly advocates transcreation as a convention-defying practice. The same transcreation practitioner [40] further argues for a radical form of specialization—hybridization, a combined skillset with transcreation as the core competence. Some of the most important names in the translation industry such as Lionbridge and TransPerfect also highlight transcreation as one of their specialties, as shown in **Figs 1 and 2**, respectively [41,42].

Please note that a colored frame has been added to the figure to distinguish the figure from its background color. The same has been done to all the following figures.

Likewise, another famous LSP [43], commonly known as the Translation Automation User Society (TAUS), considers "creative tasks as the key to translators' sustainability" (p. 8). The TAUS also provides some guidelines on transcreation [44].

International standards such as ISO:17100:2015 recognize transcreation as a value-added translation service [45].

Katan [38,46,47] has been voicing his concern over the plight of professional translators and arguing for a transcreational turn for many years. He is not alone. More and more researchers (usually also as translator trainers) are joining in the chorus for transcreation [48–

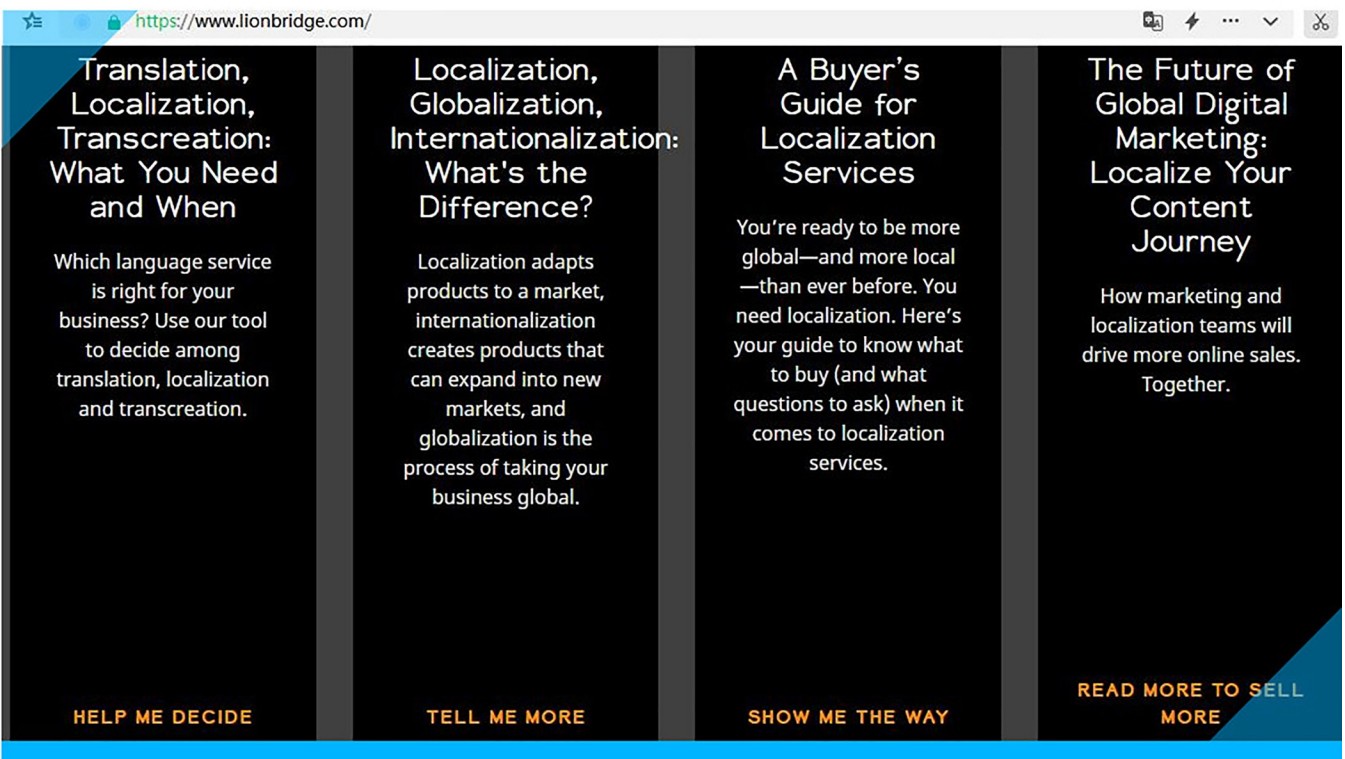

**Fig 1. Part of Lionbridge's homepage.**

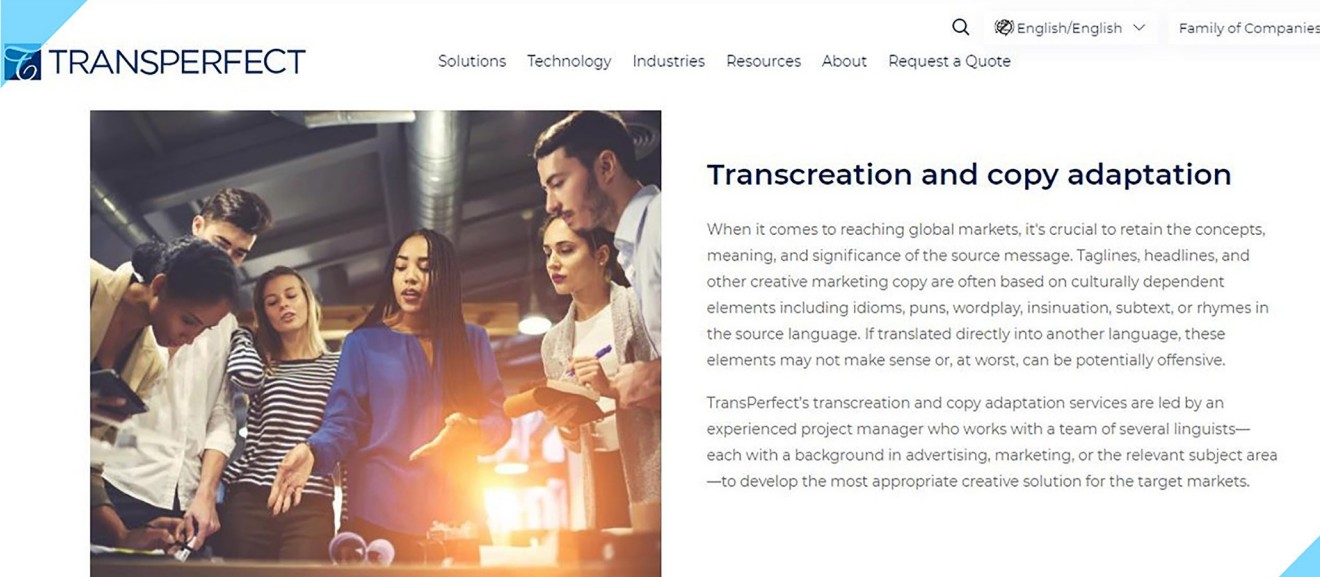

**Fig 2. Part of one of TransPerfect's webpages.**

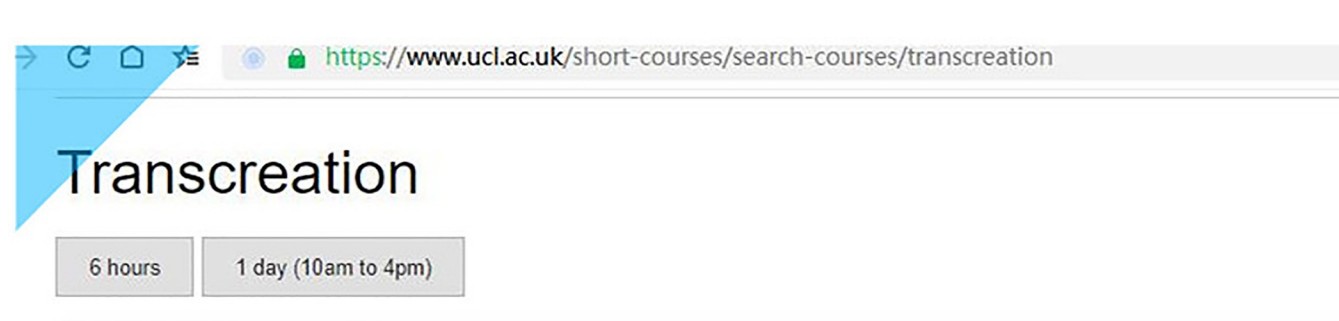

**Fig 3. UCL's short course on transcreation.**

50]. More importantly, their actions speak just as loud as their words, if not louder. Marián Morón [51] and Oliver Carreira [44] integrated transcreation into their respective translation training program at Pablo de Olavide University in Seville, Spain. Díaz-Millón et al. [52] introduced transcreation in a French-Spanish translation course at the University of Granada, Spain. Facing the threat of automation such as machine translation, Rodríguez De Céspedes [53] stressed the mastery of language skills that involve creativity although she did not point to transcreation specifically (p.113). Universities in the UK such as the University College London offer a short course, usually online, giving trainees not only an introduction to transcreation but also a chance to practice it on real texts, as presented in **Fig 3** [54].

From the above review, it can be seen that transcreation is widely recognized as one of the core competences translators should acquire. Hence the focus of this paper.

This study does not seek to make theoretical contributions but tries to give practical pedagogical advice. It does this by appealing for a shift of the focus in translator training to transcreation in general and by incorporating transcreation into the translation syllabus and translator training in particular. The goal is to improve the employability of student translators and to recommend ideas on the sustainable development in translation courses and programs in institutions of higher education. Findings show that after a one-semester trial the students have raised their awareness of transcreation as a novel approach to translation and are better able to deal with the translation of promotional materials for advertising or marketing or any other international communicative purpose.

The following sections will unfold to address the issues involved. Section 2 describes the materials and methods, especially the treatment and its impact on the transcreations efforts of students. Section 3 gives a detailed description of the survey results. Section 4 briefly discusses the findings and illustrates their implications for translation syllabus design and translator training at college. The final section draws a conclusion and gives possible lines of future research.

## Materials and methods

### Research design

Because transcreation is a core competence for translators, it is necessary to integrate transcreation into both translation syllabuses and translator training. Before the spring semester began in early 2022, this author updated a translation syllabus via a popular teaching aid, specifically a mobile app, and embedded transcreation skills in various modules of the course "Business English Translation," for the graduating class of 2023 at Guangdong Polytechnic Institute (affiliated to Guangdong Open University), or rather, two intact classes of English majors combined into one class due to lack of an adequate number of faculty members. The course lasted for the entire spring semester. Then, an online questionnaire survey was administered to the students via a popular social media mobile app called **WeChat** to obtain the overall perceptions of transcreation from the students. The experimental group ($n = 80$) is a large class for the researcher (also as course teacher) consisting of two intact classes, one with 39 students and the other with 41. A quasi-experimental pretest-posttest control group design could have been adopted. However, the transcreation skill is so important that it would have been unfair not to give the two classes the same treatment, the teaching and practicing of transcreation. Besides, separate treatments would have meant an additional workload which was difficult to handle because the faculty members were already assigned to more than enough teaching hours. Hence, the two classes were treated as one. The next best alternative could be a one-shot case study. What is a one-shot case study? "In this design, there is no control group and the subjects are given some treatment for a period of time. At the end of the period of time, the Ss [subjects] receive some sort of test on the treatment" [55] (pp. 19–20). To be more specific, the subjects in this experiment received transcreation teaching and practice for one semester after which they were asked to finish an online questionnaire survey. With the above-mentioned mobile app, the students were able to access the teaching resources including the syllabus conveniently. The syllabus specifies the competences to be acquired through this course such as this the following: "This course enables you: to translate a corporate profile from Chinese into English or vice versa, applying post-editing or transcreation; to transcreate taglines from Chinese into English or vice versa".

A flipped-classroom approach was adopted for the entire semester [56]. Assignments were given at the end of each class so that the students could watch videos and/or study lectures using Powerpoint presentations all on their own and finish the translation assignment(s) before the next class. Because the students usually submitted their translations on separate occasions, the teacher had enough time to assess the quality of the homework and sort out any problems. The next class would then focus on any problems the students encountered. Students brainstormed in groups and were tasked with developing with solutions. The teacher commented on those solutions individually and turned the student's attention to the corresponding part of the textbook or lecture documents when necessary. The students were encouraged to interrupt or raise questions whenever they had any doubts. In this way, the students were able to have a better understanding of what they dealt with. More importantly, they came to understand that different approaches could be used to address the same translation issue. Take the translation of a company profile for example. The students were given two

weeks to complete an assignment about a real furniture company. Based on the English translation of the company profile (S1 Appendix), also available at the website of the company [57], the students were asked to come up with three new different revisions or versions.

For the first revision, the students were asked to improve the original English version on their own. **Fig 4** shows one student's first revision.

Before the students prepared a second revision, the teacher made comments on their first ones and introduced them to post-editing techniques, especially transediting [58,59]. One student's second revision is provided in Fig 5.

According to Stetting [60], a translator or transeditor usually has three kinds of editing work to do: to change, to add and to remove (p.371). Changes are made for the target audience so that they can better understand the target text; in the same way, additions are made to facilitate a better understanding of the source text and related culture; words, sentences or even paragraphs are removed or deleted because they become irrelevant in the target context of the translated document (ibid).

For the third version, the teacher commented on the second revisions by the students and helped them realize that transediting might not be enough to achieve the desired promotional purpose, thus introducing them to a brand-new idea of transcreation, similar to creative writing or copywriting. **Fig 6** shows different yet quite creative versions of two students' work.

The third draft in **Fig 7** by another student is also worth citing here although it was submitted long past the due date.

The above three third versions deviate so much from the source text that it is hard to trace them back to the original text when they are translated back into the source language. According to Christiane Nord [61], they are translations or transcreations without a source text (p.17). It is no wonder that transcreation is regarded as a value-added service. For one thing, creative work does take time, as evidenced by the aforementioned three third drafts in Figs 5, 6 and 7 which were received 8, 12 and 49 days past the due date, respectively. "Creativity is the product of both nature and nurture. In some aspects it is an innate ability and in other ways it is a learned faculty that is externally acquired" [62]. The students' drafts or versions shown here reveal that transcreation helps to nurture and tap creativity on the part of the students involved. In terms of translation, the best way to bring into play one's creativity or value-added competences could be transcreation. Transcreation frees the students from trying to translate a document accurately and literally word-for-word, and so shows students that a literal translation is often not needed or can even result in undesirable meanings; it encourages the students to be creative.

## Purpose of the research

The purpose of teaching transcreation is to raise the students' awareness of transcreation and boost their employability in the translation job market. Facing similar challenges, translator trainers from other universities around the world might draw on this research to learn how to teach transcreation techniques effectively.

The research questions (RQ) are:

RQ1. How did the student translators perceive the challenges of AI or machine translation?

RQ2. How did the student translators perceive transcreation and its function in boosting their employability?

## Instruments

A 5-point Likert-scale questionnaire with 14 items was used to collect data from the 80 subjects. The questionnaire is divided into four sections. The first section with only one item is a

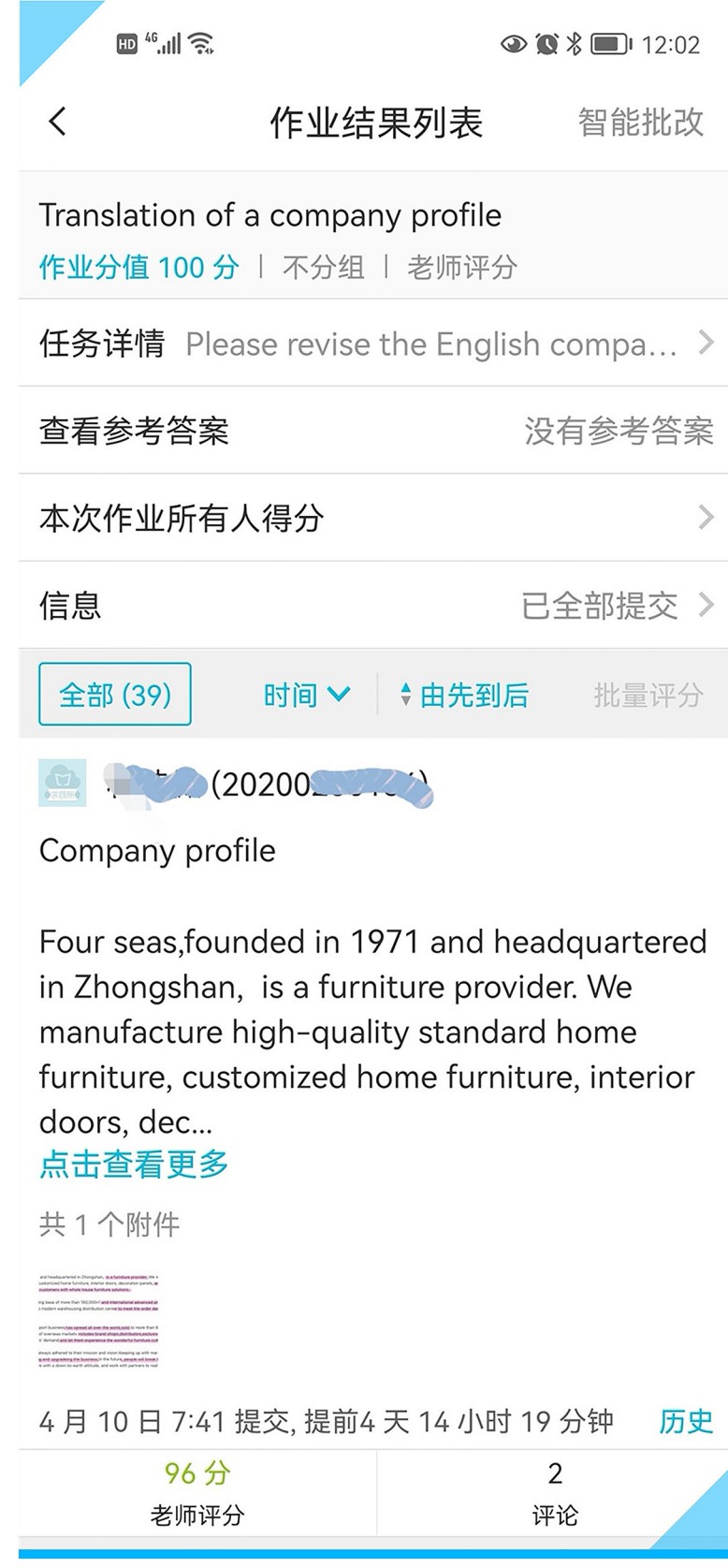

**Fig 4. One student's first revision of a furniture company profile.**

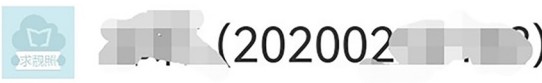

 HD 4G .ull 🛜 🌐          👁 ⏰ ✳ 🔋| 12:44

‹          作业结果列表          智能批改

求靓照     ×█ （202002██.█）

Founded in 1971, headquartered in Zhongshan, Four Seas  can offer a whole-house furnishings solution for our customers.

Over the past 50 years, Four Seas' export business has ranged to all over the world, selling to more than 60 countries and regions.

Not all furniture is borned equal.Please trust us,we will give our customers the best experience.

收起

5 月 20 日 15:53 重交，超时29 天 16 小时 53 …   历史

评分后学生将不能修改作业，你可以 允许修改

| 98 分 | 1 |
| --- | --- |
| 老师评分 | 评论 |

**Fig 5. One student's second revision of a furniture company profile.**

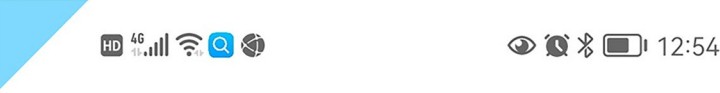

作业结果列表　　　智能批改

Four Sea furniture is famous for its simplicity and comfort. It is a luxury brand that people can afford to enhance people's happiness.
A good life comes from Four Sea furniture.
A lamp, a blanket, and a sofa allow people to fully enjoy the beauty of the day. The so-called universal home, the rearranging furniture is a way of edifying sentiment and style of life, eliminating people's life fatigue.
If you want to buy furniture, the first choice is Four Sea, we will ensure your satisfaction.
收起

5 月 20 日 19:27 重交，超时8 天 20 小时 27 …　　历史
评分后学生将不能修改作业，你可以 允许修改

| 99 分 | 3 |
| --- | --- |
| 老师评分 | 评论 |

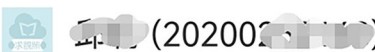 邸▇（202002▇▇▇▇）

You can imagine that you're sitting in a soft and comfortable sofa with drinking a mug of cocoa. The lights in the living room are so bright and warm. The furniture around you is so elegant and unique. What a relaxing atmosphere it is! And Four seas can provide the layout for you .
　Choose Four Seas, choose good furniture.
收起

5 月 23 日 23:23 重交，超时12 天 0 小时 23 …　　历史
评分后学生将不能修改作业，你可以 允许修改

**Fig 6. Two students' third versions of a furniture company profile.**

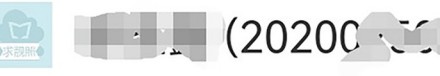

Perhaps the reason people have been living with Four Seas Furniture Company (FS) for generations is because FS is so easy to live with. Since 1971, we've been driven by a commitment to create icons of livable luxury.

 A new rug changes everything.A mirror is lighting without a bulb.Style and comfort are not mutually exclusive.Rearranging furniture is therapeutic.Rearranging art is revelatory.FS is the right place for collecting your dream furniture.

 If you don't like it, we'll take it back.

 If it makes you smile, it's your style.

**Fig 7. Another student's third version of a furniture company profile.**

statement of privacy protection and a request of the subjects' consent to the use of all the information gathered from the questionnaire for academic purposes; only those questionnaires in which the students provided informed consent are used in this study. The second section with two items elicits information on their gender and class number. The last section with one item is an open-ended question asking them for any other comments on or suggestions for this course. The other ten items included in section three are statements regarding transcreation, each with five possible responses: strongly agree, agree, undecided, disagree and strongly disagree. Those items cover four different categories: the subjects' overall impression of transcreation (one item), their perceptions of transcreation from ontological and epistemological perspectives (four items), job-related issues (four items) and an assessment of transcreation (one item).

The questionnaire was compiled via a popular free online survey website (https://www.wjx.cn/), which offers various templates to choose from. To be sure the students understood the questionnaire, a bilingual version was formulated in both English and Chinese. The questionnaire survey was administered through a popular social media mobile app called *WeChat* and was available to the students only during the last two weeks of the semester. Altogether, there were 89 submissions. That means a few respondents completed the survey more than once. Only one of the submissions with the same IP address was retained; 11 duplicate questionnaires from the students were deleted, leaving 78 unique submissions. Despite reassurances from the researcher that anonymity would be strictly kept, seven out of the 78 subjects failed to consent to the use of their information so those questionnaires are also disregarded, leaving 71 usable questionnaires (**S1 Dataset**).

## Results

### Survey overview

The 71 questionnaires are automatically processed; the results can be accessed and downloaded via the *WeChat* mini program of the free online survey software (**https://www.wjxcn/**). The statistical data are freely available in various forms such as pie and bar charts via the mini app. Fig 8 is a summary of all the data (for details, see **S2 Appendix**).

### Descriptions and interpretations of the data

Except the first three items and the last one on the questionnaire, the other ten items are concerned with transcreation, in one way or another. Based on those ten items, the following descriptions begin with Item 4, each followed by a screenshot of a hybrid presentation, both verbal and visual. The screenshot (**Fig 9**) for Item 4 is inserted here in the text. The other nine screenshots (**Figs 10**–**18**) are available at the end of the manuscript to keep them from interrupting the text.

### Item 4: This course enables me to have some idea about transcreation

The first two choices, accounting for over 90% of the responses (see **Fig 9**), indicate that the overwhelming majority of the students have benefited more or less from this course in terms of transcreation.

### Item 5: Transcreation, a form of rewriting or copywriting, deviates, to a lesser or greater extent, or even completely, from the source text, so as to better serve the target audience

What is transcreation? Answering this ontological question would involve writing a full-length paper. While academics have been busy arguing for a sound definition [5,9], most of the students, over 80% (see **Fig 10**), accept the practical definition offered here.

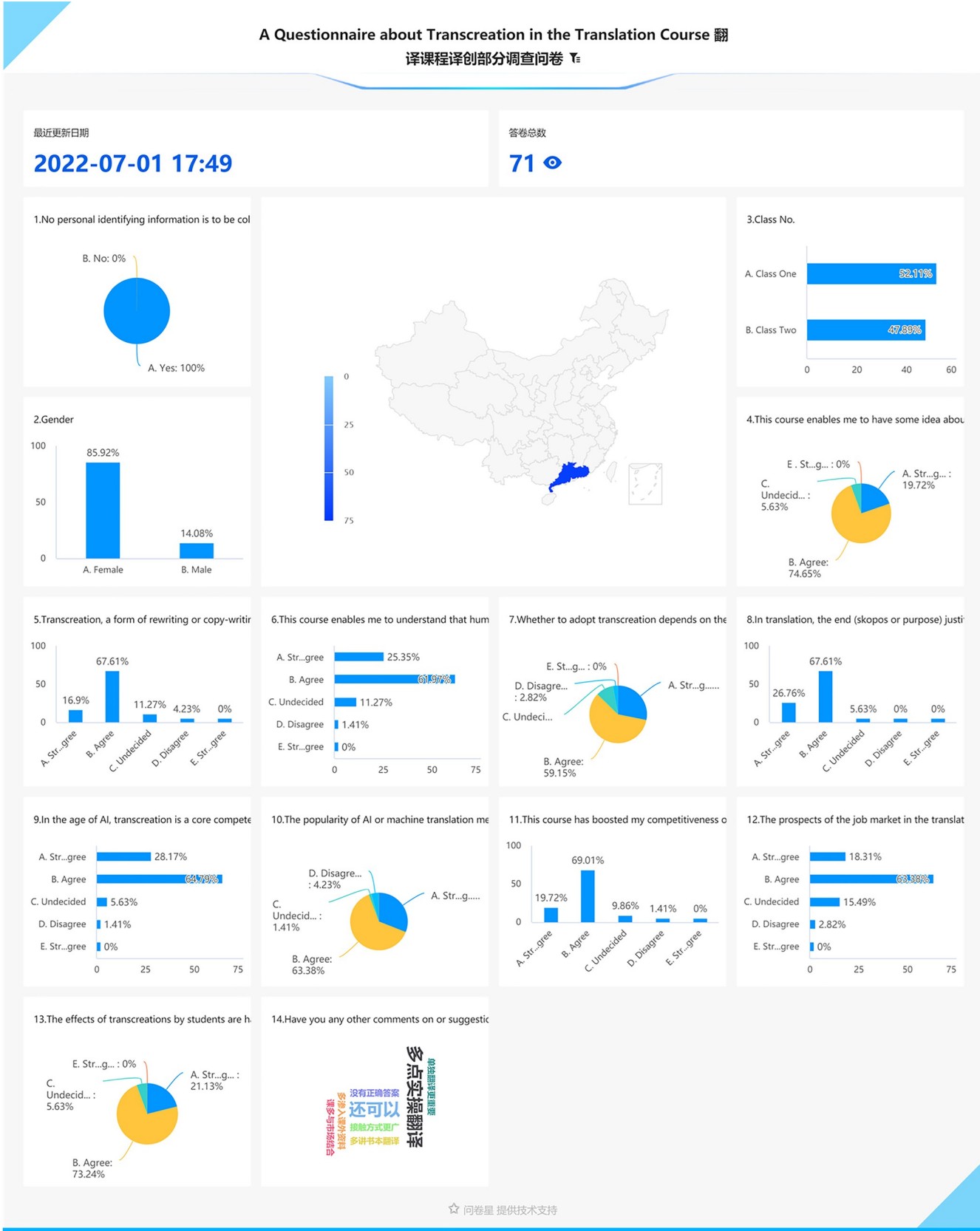

**Fig 8. Data of the questionnaire survey on a large screen.**

第4题：This course enables me to have some idea about transcreation. 这门课程让我对译创有了一些了解。  [单选题]

| 选项 | 小计 | 比例 | |
|---|---|---|---|
| A. Strongly agree | 14 | | 19.72% |
| B. Agree | 53 | | 74.65% |
| C. Undecided | 4 | | 5.63% |
| D. Disagree | 0 | | 0% |
| E . Strongly disagree | 0 | | 0% |
| 本题有效填写人次 | 71 | | |

**Fig 9. Statistical results for Item 4 on the questionnaire.**

### Item 6: This course enables me to understand that human creativity can never be replaced by AI or machine translation when it comes to cross-cultural promotional especially advertising and marketing and other communicative purposes

This understanding is crucial to translation training as more and more translation work is being taken over by AI or machine translation. Student translators just cannot help asking: "What is there left to learn from a translation course?" An awareness of the human value in

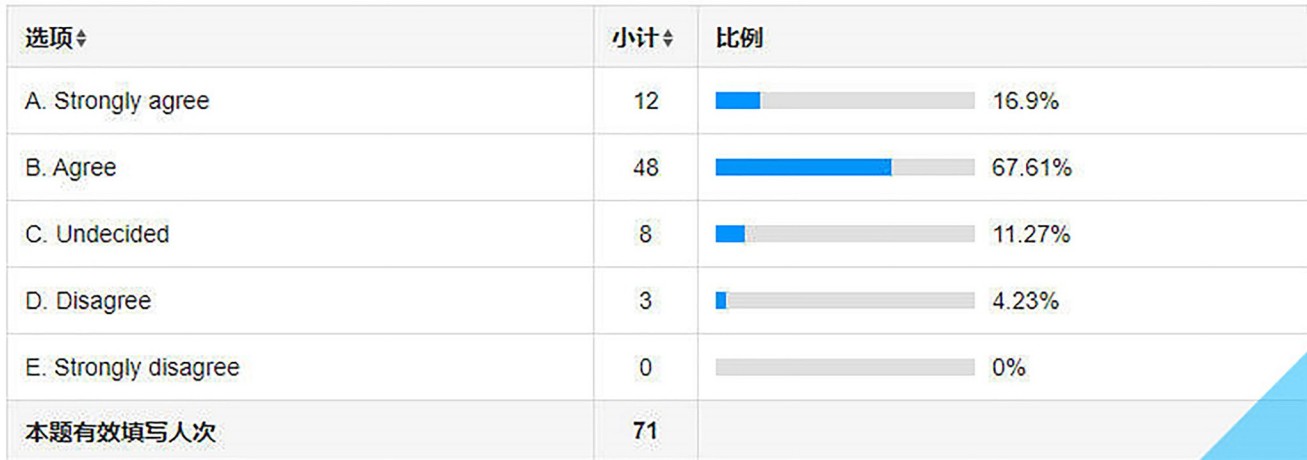

第5题：Transcreation, a form of rewriting or copy-writing, deviates, to a lesser or greater extent, or even completely, from the source text, so as to better serve the target audience. 译创，一种重写或文案创作形式，或多或少甚至完全偏离原文，以便更好地为目标读者服务。  [单选题]

| 选项 | 小计 | 比例 | |
|---|---|---|---|
| A. Strongly agree | 12 | | 16.9% |
| B. Agree | 48 | | 67.61% |
| C. Undecided | 8 | | 11.27% |
| D. Disagree | 3 | | 4.23% |
| E. Strongly disagree | 0 | | 0% |
| 本题有效填写人次 | 71 | | |

**Fig 10. Statistical results for Item 5 on the questionnaire.**

第6题: This course enables me to understand that human creativity can never be replaced by AI or machine translation when it comes to cross-cultural promotional especially advertising and marketing and other communicative purposes. 本课程让我了解到，人工智能或机器翻译在跨文化推广尤其是跨文化广告和营销以及其他国际传播方面永远无法取代人类的创造力。 [单选题]

| 选项 ♦ | 小计 ♦ | 比例 | |
|---|---|---|---|
| A. Strongly agree | 18 | | 25.35% |
| B. Agree | 44 | | 61.97% |
| C. Undecided | 8 | | 11.27% |
| D. Disagree | 1 | | 1.41% |
| E. Strongly disagree | 0 | | 0% |
| 本题有效填写人次 | 71 | | |

**Fig 11. Statistical results for Item 6 on the questionnaire.**

translation work will help boost the students' confidence not only in their current translation training at college but also in their future careers as translators and/or transcreators. Still, around 11% (see **Fig 11**) of the subjects are not sure if they could have a cutting edge over AI or machine translation.

第7题: Whether to adopt transcreation depends on the skopos or purpose of the translation involved. 是否采用译创取决于翻译的目的。 [单选题]

| 选项 ♦ | 小计 ♦ | 比例 | |
|---|---|---|---|
| A. Strongly agree | 20 | | 28.17% |
| B. Agree | 42 | | 59.15% |
| C. Undecided | 7 | | 9.86% |
| D. Disagree | 2 | | 2.82% |
| E. Strongly disagree | 0 | | 0% |
| 本题有效填写人次 | 71 | | |

**Fig 12. Statistical results for Item 7 on the questionnaire.**

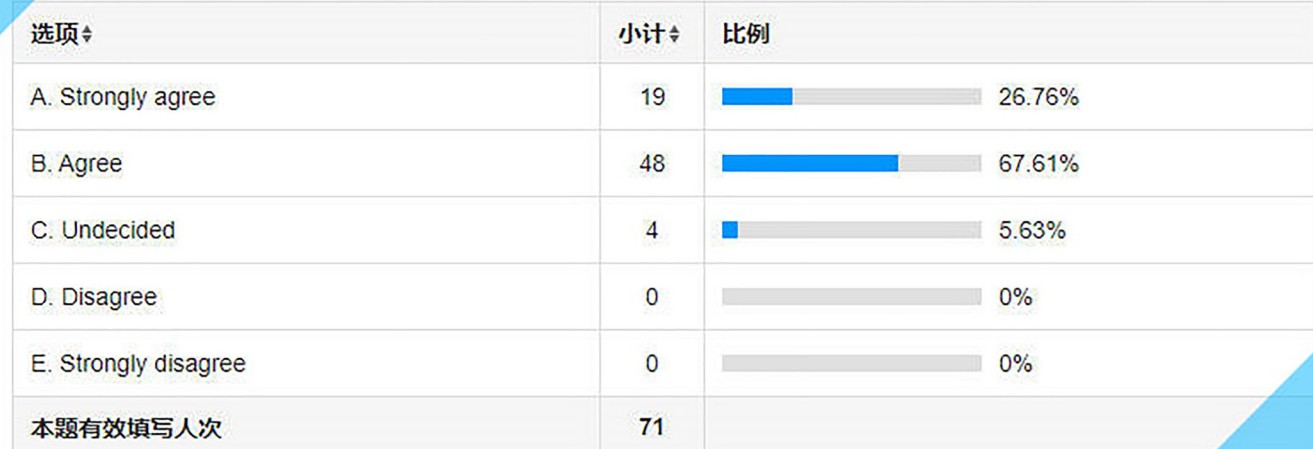

**Fig 13. Statistical results for Item 8 on the questionnaire.**

## Item 7: Whether to adopt transcreation depends on the Skopos or purpose of the translation involved

According to the Skopos theory of Hans J. Vermeer [61], a general theory of translation, "the prime principle determining any translation process is the purpose (Skopos) of the overall translational action" (p. 27). The same is true of transcreation. Most of the subjects, nearly 90% (see **Fig 12**), have no doubt about this.

第9题：In the age of AI, transcreation is a core competence or skill for translators when most of conventional translation is taken over by AI or machine translation 人工智能时代，大多数传统翻译由人工智能或机器翻译完成，译创成了译者的核心能力或技能。 [单选题]

| 选项 ⇕ | 小计 ⇕ | 比例 | |
|---|---|---|---|
| A. Strongly agree | 20 | | 28.17% |
| B. Agree | 46 | | 64.79% |
| C. Undecided | 4 | | 5.63% |
| D. Disagree | 1 | | 1.41% |
| E. Strongly disagree | 0 | | 0% |
| 本题有效填写人次 | 71 | | |

**Fig 14. Statistical results for Item 9 on the questionnaire.**

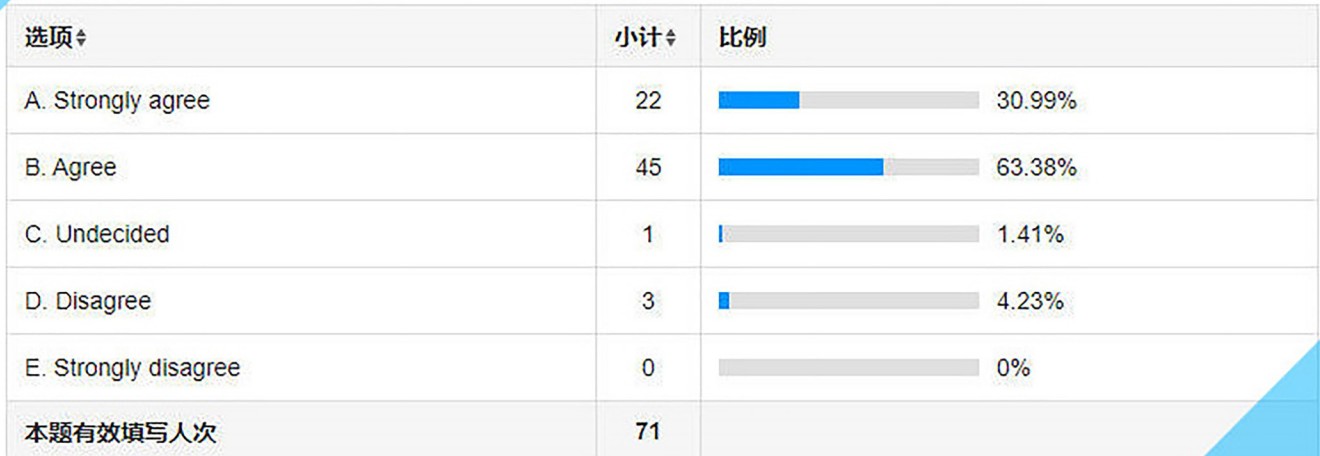

**Fig 15. Statistical results for Item 10 on the questionnaire.**

### Item 8: In translation, the end (skopos or purpose) justifies the means including but not limited to transcreation

According to Hans J. Vermeer's Skopos theory, or as Reiss and Vermeer [ibid.] put it, "the end justifies the means" as far as translation is concerned (p. 29). A misunderstanding about the skopos rule is that "a good translation should ipso facto conform or adapt to target-culture behavior or expectations" (ibid.). As a result, that "strictly excludes philological or literal or even word-for-word translations. There are many cases where relative literalism is precisely

**Fig 16. Statistical results for Item 11 on the questionnaire.**

第12题：The prospects of the job market in the translation industry are bright though facing huge challenges.翻译行业的就业市场前景光明，尽管面临巨大挑战。 [单选题]

| 选项 ÷ | 小计 ÷ | 比例 | |
|---|---|---|---|
| A. Strongly agree | 13 | | 18.31% |
| B. Agree | 45 | | 63.38% |
| C. Undecided | 11 | | 15.49% |
| D. Disagree | 2 | | 2.82% |
| E. Strongly disagree | 0 | | 0% |
| 本题有效填写人次 | 71 | | |

**Fig 17. Statistical results for Item 12 on the questionnaire.**

what the receiver (or the client or the user) needs, for example in the translation of a marriage certificate or driver's license, foreign legal texts for comparative purposes or direct quotations in newspaper reports" (ibid.). In other words, translation in general and transcreation in particular, can be justified by its skopos but the skopos rule does not necessarily trigger transcreation. It can be inferred that the vast majority of the respondents (over 94%, see **Fig 13**) have clear perceptions about the role transcreation plays in the translation process.

第13题：The effects of transcreations by students are hard to assess since the market has the final say even if translation teachers or clients are impressed. 学生的译创效果很难评估，因为即使翻译老师或客户印象深刻，最终决定权还在市场。 [单选题]

| 选项 ÷ | 小计 ÷ | 比例 | |
|---|---|---|---|
| A. Strongly agree | 15 | | 21.13% |
| B. Agree | 52 | | 73.24% |
| C. Undecided | 4 | | 5.63% |
| D. Disagree | 0 | | 0% |
| E. Strongly disagree [详细] | 0 | | 0% |
| 本题有效填写人次 | 71 | | |

**Fig 18. Statistical results for Item 13 on the questionnaire.**

### Item 9: In the age of AI, transcreation is a core competence or skill for translators when most of conventional translation is taken over by AI or machine translation

Artificial intelligence or machine translation has been encroaching on the work of translators for years. The best way to differentiate humans from machines lies in the uniqueness of humans, i.e., human creativity. Over 90% (see **Fig 14**) of the survey participants have underscored the importance of transcreation, which is closely linked to their employability if they decide to pursue a career as a translator.

### Item 10: The popularity of AI or machine translation means a great opportunity for those with skills in transcreation or copywriting

Other things being equal, those with skills in transcreation or copywriting surely have a better chance to stand out, which means more bread-winning opportunities. It is encouraging that over 94% (see **Fig 15**) are optimistic about their future as translators or copywriters.

### Item 11: This course has boosted my competitiveness or employability to some degree as a would-be translator

Although most of the respondents, slightly less than 90% (see **Fig 16**), are convinced that their competitiveness has improved, around 10% are not so sure. After all, it is one thing to realize that transcreation is important; it is quite another to be able to translate that realization into a skill. Moreover, one cannot acquire a skill at one sitting. It is especially true of transcreation, which requires constant honing.

### Item 12: The prospects of the job market in the translation industry are bright though facing huge challenges

In contrast to their responses to Item 10, we feel that the subjects seem to be sort of self-contradictory since only about 80% of them (see **Fig 17**), as opposed to more than 94% in Item 10, a difference of over 10%, are optimistic about their future careers as translators. This might be interpreted as their different understandings of the key words "opportunity" and "prospect" involved respectively in those two statements despite the fact that both words refer to a possibility. For the subjects, however, it seems that "opportunity" is only a distant possibility while "prospect" is closer to reality, especially when associated with the job market.

### Item 13: The effects of transcreations by students are hard to assess since the market has the final say even if translation teachers or clients are impressed

The overwhelming majority of the participants, over 94% (see **Fig 18**), feel that market is the one and only yardstick for the quality of a transcreation. There have been many discussions about the quality evaluation or assessment of a transcreation [63,64]. All the judgements are subjective. A transcreation of a tagline by a student, even when rated as the worst by their peers or the teacher or any other third party, might turn out to be a huge success when applied to the market. Therefore, the teacher deliberately played down the importance of assessment and encouraged the students to use their imagination in the hope that it might bring their creativity into full play when working on a transcreation.

The last item in the survey is an open-ended question, asking the subjects for further comments or suggestions. Only a few answered this question. One respondent hopes that the

course will integrate more with the translation market. As a matter of fact, most of the translation/transcreation practice materials are authentic. It would have been better if a semi-structured interview had been conducted in order to have a deeper understanding of the subjects' perceptions and opinions.

## Reliability, validity and discriminability

According to D.A. de Vaus [65], three very important principles should be followed when designing survey questions: reliability, validity and question discrimination (currently called discriminability) (p.96). Therefore, three statistical tests for reliability, validity and discriminability were conducted to ensure that the survey results are trustworthy. Logging onto the *WeChat Web*, a web version of the mobile app, and clicking on the mini program of the free online survey software (https://www.wjx.cn/) will allow the user to conduct an analysis of these three factors. Statistical analyses such as reliability analysis can be conducted via a third-party program, namely, SPSSAU (also available at https://www.spssau.com), an online software similar to Statistical Product and Service Solutions (SPSS). First, we conducted an analysis of this survey's reliability using the English edition. On the left side of the webpage, one can click on "**Reliability**." In the middle, the ten items covered in this survey can be chosen, after which they will be moved from the middle to the right (**Fig 19**).

Clicking on "**Start analysis**" instantly makes the results available as presented in the screenshot (see **Fig 20**).

The analysis results can then be exported in different formats (Excel, PDF and Word). **Fig 21** shows a screenshot of the exported reliability analysis results (**S1 File**).

Next, we had to check each item to see if it really fits the scale. This process actually involves two aspects—unidimensionality and reliability. A unidimensional scale is one where each item measures the same underlying concept. Items that do not measure the concept should be deleted. We should check to see if the responses to a particular item reflect the pattern of

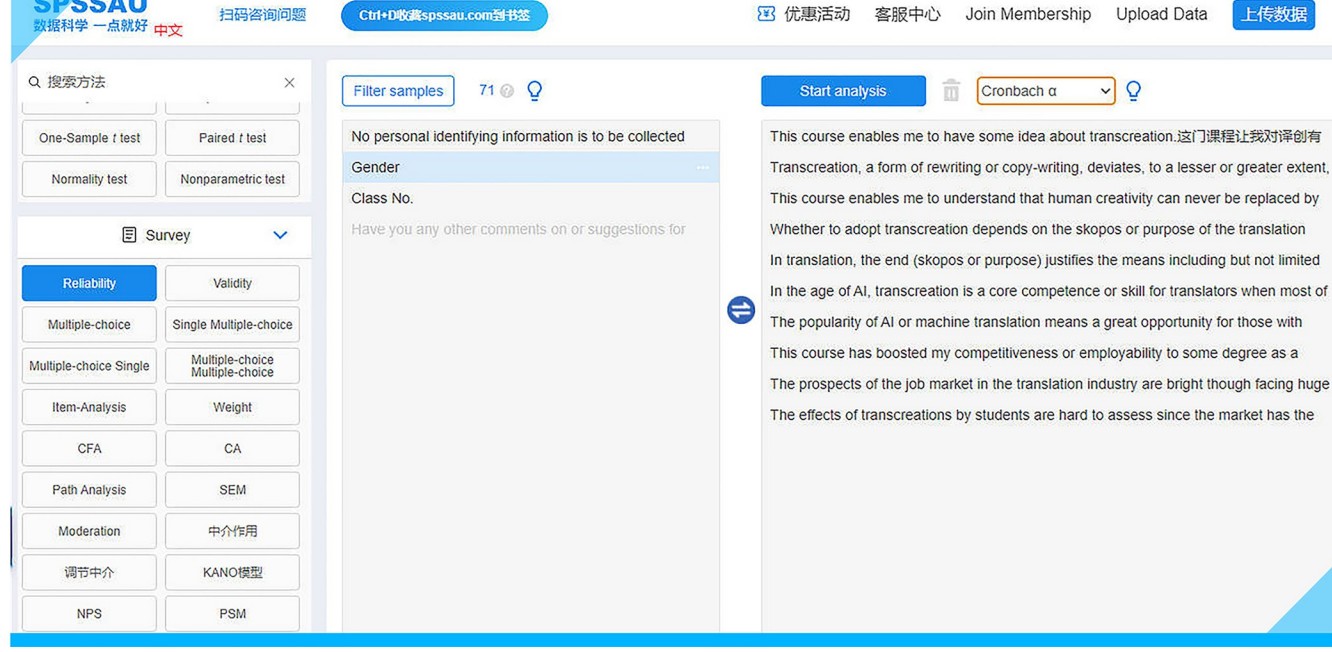

**Fig 19. The SPSSAU interface, an online software similar to SPSS.**

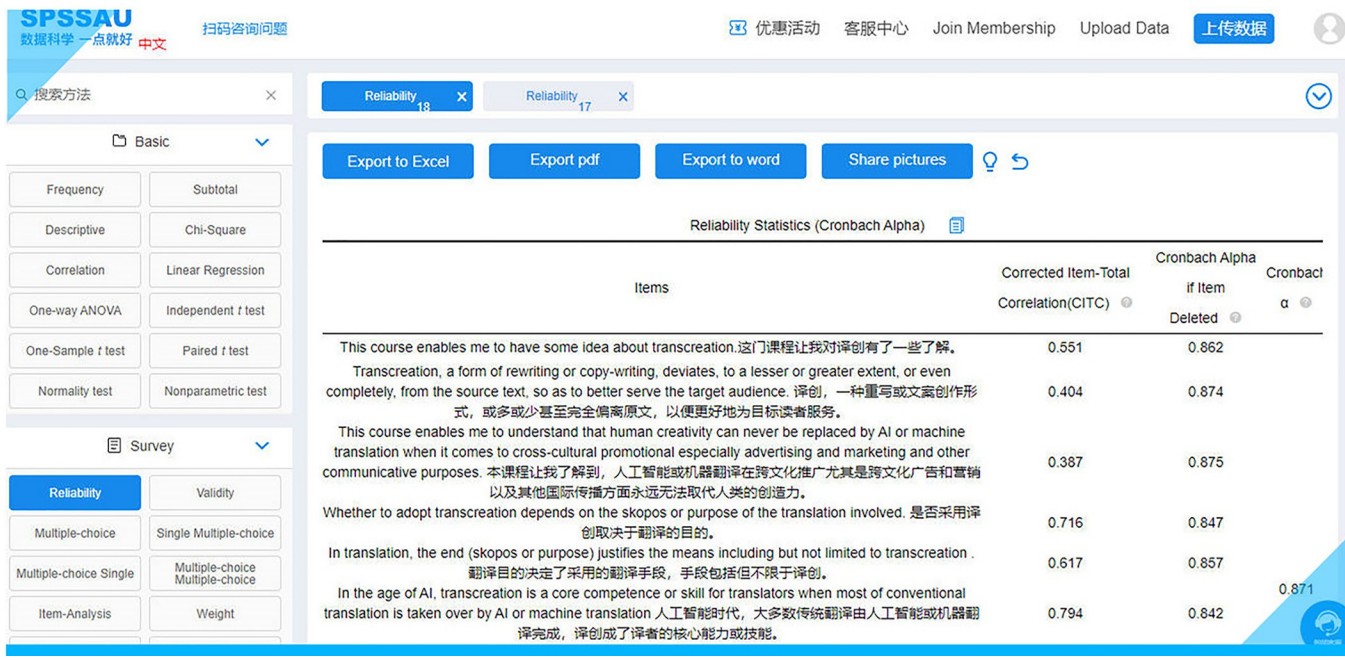

**Fig 20. Reliability analysis results via SPSSAU.**

responses on other items. The best way to do this is to calculate a correlation coefficient between response on each item with the students' responses on the remaining items. Correlation coefficients range between 0 and 1. The coefficient that tests the fit between an item and the rest of the scale is called an item-to-scale coefficient or an item-to-total coefficient (ibid.: 184). The higher it is, the better. If it is less than 0.3, then the item should be dropped from the scale. As can be seen from **Fig 21**, all ten 10 items have an item-to-total correlation of over 0.3.

As for reliability, a good way to test reliability is to "look at the consistency of a person's response on an item compared to each other scale item (item–item correlations) (ibid.). In this way, the overall reliability of the scale can be measured. The index of this is given by a statistic Cronbach's alpha coefficient which ranges between 0 and 1" (ibid.). The alpha should be at least 0.7. The higher it is, the better. **Fig 21** shows that the deletion of any item has little effect on the alpha of the scale since it always stays above 0.7. The Cronbach's alpha (= 0.871) indicates that all the items in this survey have a high level of internal consistency and thus can be deemed reliable.

In the same way, the validity statistics were derived. **Fig 22** is a screenshot of the exported validity analysis results (**S2 File**).

How can the validity of a scale be established? According to D.A. de Vaus(ibid.), "[f]actor analysis is an appropriate method for scale development when analysing a set of interval-level, non-dichotomous variables. It is a mathematically complex method of reducing a large set of variables to a smaller set of underlying variables referred to as factors" (p.186). With the aid of computer packages such as SPSS or SPSSAU, complex computations can be easily handled. If the Kaiser-Meyer-Olkin statistic (called the KMO statistic) is above 0.7, then the correlations, on the whole, are high enough to validate factor analysis (ibid.). In this scale, the value of KMO (= 0.827) is greater than 0.7, suggesting that the data gathered from this survey were valid.

Another measure of the trustworthiness of a scale is discriminability, formerly called question discrimination, which refers to the "capacity of a question to distinguish between cases

| Reliability Statistics (Cronbach Alpha) | | | |
|---|---|---|---|
| Items | Corrected Item-Total Correlation(CITC) | Cronbach Alpha if Item Deleted | Cronbach α |
| 4、 This course enables me to have some idea about transcreation. | 0.551 | 0.862 | |
| 5、 Transcreation, a form of rewriting or copy-writing, deviates, to a lesser or greater extent, or even completely, from the source text, so as to better serve the target audience. | 0.404 | 0.874 | |
| 6、 This course enables me to understand that human creativity can never be replaced by AI or machine translation when it comes to cross-cultural promotional especially advertising and marketing and other communicative purposes. | 0.387 | 0.875 | |
| 7、 Whether to adopt transcreation depends on the skopos or purpose of the translation involved. | 0.716 | 0.847 | |
| 8、 In translation, the end (skopos or purpose) justifies the means including but not limited to transcreation . | 0.617 | 0.857 | |
| 9、 In the age of AI, transcreation is a core competence or skill for translators when most of conventional translation is taken over by AI or machine translation | 0.794 | 0.842 | 0.871 |
| 10、 The popularity of AI or machine translation means a great opportunity for those with skills in transcreation or copy-writing. | 0.595 | 0.858 | |
| 11、 This course has boosted my competitiveness or employability to some degree as a would-be translator. | 0.659 | 0.853 | |
| 12、 The prospects of the job market in the translation industry are bright though facing huge challenges. | 0.612 | 0.857 | |
| 13、 The effects of transcreations by students are hard to assess since the market has the final say even if translation teachers or clients are impressed. | 0.637 | 0.856 | |

Cronbach α (Standardized): 0.875

**Fig 21. Exported reliability analysis results via SPSSAU.**

where real differences in the item exist. A question with good discrimination will be sensitive to real differences" (ibid.:363). Discriminability quantifies the degree to which individual items are relatively similar to one another [66]. A discriminability analysis (called "item-analysis" by SPSSAU) was made of the ten items. **Table 1** shows that for each of the ten items, a significant difference exists between the low- and the high-score groups since $p$ is less than 0.05, which means each item has a high level of discriminability. To save space, a corresponding number was substituted for the content of each of the ten items.

For details, please see the exported discriminability (item-analysis) document via SPSSAU (**S1 Table**).

The three statistical tests done above reaffirm the trustworthiness of the survey results.

## Discussion

Before starting this discussion, it is necessary to look at two research questions (RQ) again:

RQ1. How did the student translators perceive the challenges of AI or machine translation?

RQ2. How did the student translators perceive transcreation and its function in improving their employability?

The survey findings show that the students have raised their awareness of transcreation as a novel approach to translation and according to their responses, especially to Items 6, 9 and 10, most of them would perceive the challenges of AI or machine translation not as a threat but as an opportunity to tap into the potentials of their own creativity; at the same time, the students regard transcreation as a core competence or a pillar to sustain their future careers as translators. It can be inferred from their responses to Item 11 that with the exception of around 10% of the students, the respondents would agree that transcreation will play a role in improving

| Validity Analysis | Factor Loadings | | Communalities |
|---|---|---|---|
| Items | Factor 1 | Factor 2 | |
| 4. This course enables me to have some idea about transcreation. | 0.253 | 0.825 | 0.745 |
| 5. Transcreation, a form of rewriting or copy-writing, deviates, to a lesser or greater extent, or even completely, from the source text, so as to better serve the target audience. | 0.124 | 0.758 | 0.590 |
| 6. This course enables me to understand that human creativity can never be replaced by AI or machine translation when it comes to cross-cultural promotional especially advertising and marketing and other communicative purposes. | 0.160 | 0.659 | 0.460 |
| 7. Whether to adopt transcreation depends on the skopos or purpose of the translation involved. | 0.857 | 0.118 | 0.749 |
| 8. In translation, the end (skopos or purpose) justifies the means including but not limited to transcreation . | 0.683 | 0.215 | 0.512 |
| 9. In the age of AI, transcreation is a core competence or skill for translators when most of conventional translation is taken over by AI or machine translation | 0.798 | 0.366 | 0.770 |
| 10. The popularity of AI or machine translation means a great opportunity for those with skills in transcreation or copy-writing. | 0.785 | 0.019 | 0.617 |
| 11. This course has boosted my competitiveness or employability to some degree as a would-be translator. | 0.685 | 0.316 | 0.569 |
| 12. The prospects of the job market in the translation industry are bright though facing huge challenges. | 0.720 | 0.145 | 0.540 |
| 13. The effects of transcreations by students are hard to assess since the market has the final say even if translation teachers or clients are impressed. | 0.662 | 0.346 | 0.557 |
| Eigenvalues (Initial) | 4.830 | 1.278 | - |
| % of Variance (Initial) | 48.297% | 12.782% | - |
| % of Cum. Variance (Initial) | 48.297% | 61.080% | - |
| Eigenvalues (Rotated) | 3.984 | 2.124 | - |
| % of Variance (Rotated) | 39.839% | 21.241% | - |
| % of Cum. Variance (Rotated) | 39.839% | 61.080% | - |
| KMO | 0.827 | | - |
| Bartlett's Test of Sphericity (Chi-Square) | 336.525 | | - |
| df | 45 | | - |
| p value | 0.000 | | - |

Note: Blue indicates that the absolute value of loading is greater than 0.4, and red indicates that the communality is less than 0.4.

**Fig 22. Exported validity analysis results via SPSSAU.**

**Table 1. Item-analysis.**

| | Group (M±SD) | | T (CR) | p |
|---|---|---|---|---|
| | Low Scores (n = 20) | High Scores (n = 42) | | |
| 1 | 1.40±0.50 | 2.05±0.31 | 5.306 | 0.000** |
| 2 | 1.60±0.75 | 2.17±0.49 | 3.067 | 0.005** |
| 3 | 1.50±0.61 | 2.07±0.51 | 3.637 | 0.001** |
| 4 | 1.25±0.44 | 2.24±0.58 | 6.76 | 0.000** |
| 5 | 1.30±0.47 | 2.07±0.34 | 6.559 | 0.000** |
| 6 | 1.05±0.22 | 2.14±0.42 | 10.953 | 0.000** |
| 7 | 1.25±0.55 | 2.12±0.55 | 5.815 | 0.000** |
| 8 | 1.40±0.60 | 2.17±0.44 | 5.118 | 0.000** |
| 9 | 1.45±0.60 | 2.31±0.56 | 5.49 | 0.000** |
| 10 | 1.35±0.49 | 2.07±0.26 | 6.188 | 0.000** |

their employability. These findings are in line with those made by Marián Morón [51] who implemented career-oriented transcreation training for the students at the University Pablo de Olavide in Seville, Spain. However, different aspects are highlighted in the present research; a shift of focus was made to transcreation as a core competence in translator training and in the embedding of transcreation in the syllabus design. What is novel about the present study lies in these highlights, which constitute its pedagogical contributions to translator training. The implications are at least two-fold. First, it is high time for translation programs at colleges and universities around the globe to shift their focus to transcreation as a core competence in translation teaching and translator training. Secondly, translation syllabus designs need to be updated and include transcreation as a core component or a core module. People might ask, "Why all this fuss about transcreation?" Insights from cross-cultural psychology can inform translation in general and transcreation in particular. Translation or transcreation is an activity within intercultural or cross-cultural communication. According to Naji Abi-Hashem and Christa E. Peterson [67], "[i]ntercultural communication is an exchange of information between two or more people of different ethnic groups, races, cultures, or nationalities" and the "process involves learning the proper knowledge, skills, and attitudes to navigate cultural differences with sensitivity, creativity, and flexibility—and most importantly to avoid prejudging others who are socially or ethnically different." According to Van der Zee and Van Oudenhoven [68], to achieve effective intercultural or cross-cultural communication requires five personality traits: cultural empathy, open-mindedness, social initiative, emotional stability, and flexibility. Cultural empathy refers to empathizing with the feelings, thoughts, and behaviors of individuals from a different culture while open-mindedness points to an open and unprejudiced attitude toward cultural differences; social initiative is a tendency to actively approach social situations while emotional stability refers to an ability to stay calm under novel and stressful conditions; the fifth trait, flexibility, involves interpreting novel situations as a positive challenge and adapting to these situations accordingly (ibid.).

In light of these findings, it would be helpful to take a quick look at some of the university programs in line with the European Master's in Translation (EMT) standards. The Directorate-General for Translation (DGT) of the European Commission is responsible for setting up an EMT network. "European Master's in Translation (EMT) is a quality label for MA [Master of Arts] university programs in translation. The DGT awards it to higher education programs that meet agreed professional standards and market demands" [69]. It would be very interesting to compare EMT programs at all the universities across the European continent in terms of transcreation, which, however, is not the focus of this paper. To illustrate whether transcreation is covered at institutions of higher education, we only selected UK study programs, which involved thirteen universities and "offer high quality master's level training for translators in line with the EMT standards, although following Brexit and the withdrawal of the UK from Erasmus+, they are no longer members of the EMT network 2019–2024" [70]. A search of the webpages of those universities for the keyword "transcreation" produced no matching results except for four universities (**Table 2**).

It can be inferred that a prevailingly weak awareness of transcreation exists among most institutions of higher education in the UK. Therefore, the findings of this research could be used as a wake-up call for MA university programs in translation that are not quite ready to meet the challenges of AI or machine translation.

## Conclusion

In conclusion, the limitations of a one-shot case study are obvious with no random sampling or control group. The focus of this research is not to generalize the findings to the whole

**Table 2. Search results for the keyword "transcreation".**

| Universities | Matching Results | |
|---|---|---|
| | Yes | No |
| **Bath:** University of Bath | | √ |
| **Birmingham:** Aston University | | √ |
| **Durham:** Durham University | | √ |
| **Edinburgh:** Heriot-Watt University | | √ |
| **Guildford:** University of Surrey | √ | |
| **Leeds:** University of Leeds | √ | |
| **London:** University of Roehampton | √ | |
| **London:** University of Westminster | | √ |
| **London:** London Metropolitan University | √ | |
| **Newcastle Upon Tyne:** Newcastle University | | √ |
| **Portsmouth:** University of Portsmouth | | √ |
| **Swansea:** Swansea University | | √ |
| **Sheffield:** The University of Sheffield | | √ |

population but rather, to deepen our understanding of the particular case involved and come up with a solution to the challenges posed by AI or machine translation, which greatly threatens the sustainability of not merely university programs in translation but the translation industry as a whole. This paper contributes in two ways: theoretically, it proposes a new definition of transcreation although no further elaborations have been made; on the practical side, it appeals for a shift of the focus in translator training to transcreation and incorporating transcreation into the translation syllabus and translator training so that the employability of student translators can be boosted while sustainable development in translation courses and programs can be maintained in institutions of higher education. Future research may explore how best to design a translation syllabus and optimize the configuration of different modules involving transcreation. Transcreation without a source text can also be probed in the future.

## Supporting information

**S1 Table. Exported discriminability (item-analysis) document via SPSSAU.**
(PDF)

**S1 Appendix. The English translation of the furniture company profile available at the website.**
(PDF)

**S2 Appendix. The questionnaire survey statistics in verbal and visual forms.**
(PDF)

**S1 Dataset. Questionnaire survey data.**
(XLSX)

**S1 File. Exported reliability analysis results via SPSSAU.**
(PDF)

**S2 File. Exported validity analysis results via SPSSAU.**
(PDF)

## Acknowledgments

I thank Philip E. Hyatt for improving the use of English in the manuscript.

## Author Contributions

**Conceptualization:** Minghai Zhu.

**Data curation:** Minghai Zhu.

**Formal analysis:** Minghai Zhu.

**Funding acquisition:** Minghai Zhu.

**Investigation:** Minghai Zhu.

**Methodology:** Minghai Zhu.

**Project administration:** Minghai Zhu.

**Resources:** Minghai Zhu.

**Software:** Minghai Zhu.

**Supervision:** Minghai Zhu.

**Validation:** Minghai Zhu.

**Visualization:** Minghai Zhu.

**Writing – original draft:** Minghai Zhu.

**Writing – review & editing:** Minghai Zhu.

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
