## [Decision Letter · Decision Letter 0]

12 Jan 2023

PONE-D-22-30289Sustainability of Translator Training in Higher EducationPLOS ONE

Dear Dr. Zhu,

Thank you for submitting your manuscript to PLOS ONE. After careful consideration, we feel that it has merit but does not fully meet PLOS ONE’s publication criteria as it currently stands. Therefore, we invite you to submit a revised version of the manuscript that addresses the points raised during the review process.

We look forward to receiving your revised manuscript.

Kind regards,

Grant Rich, Ph.D.

Academic Editor

PLOS ONE

Journal Requirements:

3. We note that Figure 2 includes an image of a participant in the study.

Additional Editor Comments:

Your submission is recommended for major revision, then resubmit.

I advise carefully editing the article for English language writing and fluency.

Also be sure to define transcreation in the first page or two of the article; this word is very unusual in English.

Please check you calculations and use of validity factor analysis and Chronbach's alpha with a statistician to confirm it is appropriate and correct; there should be more explanation about what statistical tests were used in figures 22 and 23 and you think they are the appropriate tests and what the reasons were to conduct them

I paste the two reviewers’ comments below

REVIEWER ONE SAYS “Major revision” needed and says

“Well, with all respect, this manuscript does not read with a flow, the sequence of ideas and themes is scattered/disrupted, the English language seems in cohesive, and development and rational of arguments is not well defined.

For more detailed review, please see Attached File for my extended comments” [ The detailed review is

This is good effort on the part of the authors and seems to be timely as globalization and digitalization are taking place rapidly.

However, the Abstract and Introduction sections are not easy to read.

Not flowing structure and content. Authors dive into the subjects w/o adequate definition of terms or gradual building the ground stage

Opening paragraph is too technical for the general readers

Should start like : The UN has developed 17 Sustainable Development Goal (SDG) due to the need of … and the new situation…. (before mentioning target 4.4 or such..)

Again, the opening sentence/paragraph of the Abstract is way too long and complex. Those of us who are not familiar with translators and transcreation, these themes need to be introduced and clarified.

38. To achieve the sustainable development goal in education set by the United Nations for

39. 40 2030, one of whose targets is to substantially increase the number of youth and adults possessing

40. 41 relevant skills, including technical and vocational skills, for employment, decent jobs and entrepre-

41. 42 neurship, students need to be equipped with the core competences identified for the field they spe-

42. 43 cialize in. For student translators, transcreation is a core competence they are expected to acquire.

My Suggestion:

The United Nations has set a Sustainable Development Goal in education to be met hopefully by 2030 (SDG). One of the target areas is to increase, in a substantial way, the number of youth and adults possessing relevant trainings and proficiencies, including technical and vocational skills for employment, affordable jobs, and decent entrepreneurships. Enrolled students need to be equipped with core competencies suitable for the fields in which they are specializing. Translation is one of these fields of specialty. Thus, for student translators, “transcreation” is a core competency they are expected to acquire and practice.

For example, I had to look up the word transcreation online to exactly understand what the authors mean by it and what are they talking about!?! A brief definition of terminologies is essential to any scholarly document before engaging in any detailed discussion.

This is what I found from Google:

Transcreation is a fusion of the words “translation” and “creation.”

It describes copywriting content in a source text that needs to be made coherent, relevant, etc. in a new language. Sometimes transcreation is also called “creative translation.” Namely, because the content isn't translated word for word.

The Conclusion has more flow and clear insight than the opening & the introduction. I suggest the authors include some of that clear description in this paper early on so the educated reader can have an idea of the purpose and direction of this manuscript.

481. 481 In conclusion, the limitations of a one-shot case study are obvious with no random

482. 482 sampling nor a control group. The focus is not to generalize the findings to the whole

483. 483 population but rather, to deepen our understanding of the particular case involved and

484. 484 come up with a solution to the challenges posed by AI and machine translation, which

485. 485 greatly threaten the sustainability of not merely university programs in translation but the

486. 486 translation industry as a whole. Therefore, the ultimate purpose of this paper, in addition

487. 487 to shedding some light on future research along this line, is to appeal for a shift of the

488. 488 focus in translator training to transcreation so that the employability of student translators

489. 489 can be boosted on the one hand and sustainable development in translation courses and

490. 490 programs can be maintained in institutions of higher education on the other. Future re-

491. 491 search may explore how best to design a translation syllabus and optimize the configura-

492. 492 tion of different modules involving transcreation. Transcreation without a source text can

493. 493 also be probed.

Also seems to me that the author’s mother language is not American-English, thus the discrepancies in the text and lack of flow and connectivity. So, the manuscript can benefit from revision to be made by a couple of English speaking educators who are also familiar with the subject matter.

Too many comas in the text ,,,,,,,, making sentences bit fragmented.

The term Student Translator perhaps should be hyphenated for better reading and conceptualization of this specific target population: student-translator(s) or trainer-translators

Text will better read if authors start some paragraphs with “According to so & so (year), … Rather plugging that phrase at the very end of the sentence, SEE line 79 & 106

The statement of the problem is not clear! Because there are online Apps or engines that engage in translation from local-national into global or international languages, does not necessarily eliminate the need for the human factors (translators) like any other discipline or market item, available in person in tangible forms and also electronically in online forms.

Maybe this very paper has been translated into English with the help of computer software program or similar engine : )

The paper quickly moves to become a Case Study - not a discussion/analysis of a major theme topic. Then moves to talk about EDITING on top of the primary subject Translating.

The research method and survey, followed by statistics, then interpretations, all seem reasonable sequences and important steps.

The paper warns against literal translation word for word, verbatim, which is extremely important and commendable.

Then the authors bring up the cross-cultural issues and dynamics, which are extremely important and organic part of any interaction, relationship, and communication: verbal or non-verbal, overt or covert, explicit or implicit.

Here the paper will benefit from citing some major definitions of CULTURE, citing some insights/works in the fields of:

cross-cultural human services,

intercultural & intracultural communication,

layers of multiculturalism,

psychology/sociology of globalization,

links of language-brain-culture,

mentality and tradition,

norms and customs,

social meaning-making,

religion and worldview,

Cultural sensitivity vs counter-culture approaches,

and the like….

Also some literature review needed re: human-machine interaction, experts skills & productivity versus smart machines & computers abilities/productivity. There are a lot of research and insights there.

In the discussion section, it is not clear why the authors focused only on some universities in UK to check whether their MA program incorporates Transcreation, to find only 4 of them do. Since the DGT is the credentialing agency for the EMT in Europe, why not compare that in the whole continent,

It is also common sense that we as trained human students or accomplished experts/scholars alike are aware of the new competition and subtle abilities of computers-soft wares and apps – so we adjust, learn to maneuver, and work in parallel rather one eliminates the other. SO, what is novel about this study and it s finding or recommendation? Not clear!

Some technical notes:

Text needs revision for better language, structure, flow, and cleaning of punctuations.

The tables and inserts within the text are way too many, so I am not sure how professional to keep so many images interrupting the text. Otherwise they need to be reorganized in better way.

The Reference section /Bibliography is too short for such a compound subject. Definitely, it is not in APA style yet looks bit crowded.

SECOND REVIEWER SAID MINOR REVISION and comments below

“The article needs minor revision.It needs to focus on cultural and political discourse of education and post colonial movement. The article lacks theoretical grounds that focus on the discursive parlance of education and its implications for marginal versus core meanings”

Reviewers' comments:

Reviewer's Responses to Questions

**Comments to the Author**

1. Is the manuscript technically sound, and do the data support the conclusions?

Reviewer #1: Partly

Reviewer #2: Partly

2. Has the statistical analysis been performed appropriately and rigorously? 

Reviewer #1: I Don't Know

Reviewer #2: Yes

3. Have the authors made all data underlying the findings in their manuscript fully available?

Reviewer #1: Yes

Reviewer #2: Yes

4. Is the manuscript presented in an intelligible fashion and written in standard English?

Reviewer #1: No

Reviewer #2: Yes

5. Review Comments to the Author

Reviewer #1: Well, with all respect, this manuscript does not read with a flow, the sequence of ideas and themes is scattered/disrupted, the English language seems in cohesive, and development and rational of arguments is not well defined.

For more detailed review, please see Attached File for my extended comments

Reviewer #2: The article needs minor revision.It needs to focus on cultural and political discourse of education and post colonial movement. The article lacks theoretical grounds that focus on the discursive parlance of education and its implications for marginal versus core meanings.

6. PLOS authors have the option to publish the peer review history of their article (what does this mean?). If published, this will include your full peer review and any attached files.

Reviewer #1: **Yes: **Naji Abi-Hashem, MDiv, MA, PHD, DAAETS

Reviewer #2: **Yes: **Dr. Sayyed Mohsen Fatemi, Ph.D.

----

Grant J. Rich, PhD LMT BCTMBPresident-Elect Society for Peace, Conflict, and Violence (APA D48)President-Elect Society for Media Psychology and Technology (APA D46)

Fellow, Association for Psychological Science (APS)Fellow, American Psychological Association (APA)Senior Contributing Faculty, Walden UniversityDr. Rich's SPN Website: http://rich.socialpsychology.org/**Book Website** (Rich, Gielen, & Takooshian, 2017)http://www.infoagepub.com/products/Internationalizing-the-Teaching-of-Psychology**Book Website** (Rich & Sirikantraporn, 2018)https://rowman.com/ISBN/9781498554831/Human-Strengths-and-Resilience-Cross-Cultural-and-International-Perspectives#**Book Website** (Rich, Jaafar, & Barron, 2020) Psychology in Southeast Asia. Routledge.https://www.routledge.com/Psychology-in-Southeast-Asia-Sociocultural-Clinical-and-Health-Perspectives/Rich-Jaafar-Barron/p/book/9780367492144**Book Website** (Rich & Ramkumar, 2022) Psychology in Oceania and the Caribbean, Springerhttps://link.springer.com/book/10.1007/978-3-030-87763-7#editorsandaffiliations **Book Website**(Rich, Kuriansky, Gielen, & Kaplan, in press) * Psychosocial Experiences and Adjustment of Migrants: Coming to the USA, Elsevier**https://www.elsevier.com/books/psychosocial-experiences-and-adjustment-of-migrants/rich/978-0-12-823794-6* **Book **(Rich, Kumar, & Farley, in contract)* Handbook of **Media Psychology and Technology-The Science and the Practice,** Springer*

---

## [Author Response · Author response to Decision Letter 0]

28 Feb 2023

Dear Dr. Grant Rich,

Thank you and the two reviewers Dr. Naji Abi-Hashem and Dr. Sayyed Mohsen Fatemi so much for your comments and suggestions. Due to your help, the revised manuscript hopefully has been substantially improved. I will respond to each point raised by you all in three parts as follows: Part 1: Response to the Academic Editor; Part 2: Response to the First Reviewer; Part 3: Response to the Second Reviewer. 

Part 1: Response to the Academic Editor, Dr. Grant Rich

Q1. Please ensure that your manuscript meets PLOS ONE's style requirements, including those for file naming.

A1: The style requirements have now been met in the revised manuscript. And the files have been renamed accordingly.

Q2. In your Data Availability statement, you have not specified where the minimal data set underlying the results described in your manuscript can be found.

A2: The dataset has now been specified in the Data Availability statement as well as in the manuscript and uploaded as a Supporting Information file, as shown in the manuscript:

428 leaving 71 usable questionnaires (S1 Dataset)

759 S1 Dataset. Questionnaire survey data.

Q3. We note that Figure 2 includes an image of a participant in the study.

A3: Figure 2 is actually a screenshot of part of a publicly accessible webpage of a language service provider called TransPerfect, whose link is given below, as cited in the references of the manuscript. So, it has nothing to do with the participants in the study:

42. Transperfect. Multicultural Marketing Services：TransPerfect [Internet]. [cited 2023 Feb 8]. Available from: https://www.transperfect.com/solutions/global-brand-management/multicultural-marketing

Q4. Additional Editor Comments

Q4.1. Your submission is recommended for major revision, then resubmit.

A4.1: Major revisions have been made upon your advice and that of the first reviewer Dr. Naji Abi-Hashem who provided me with a lot of references after I contacted him by email. Dr. Sayyed Mohsen Fatemi has also mentioned a lot of references by email, which, personally, are more relevant to literary translation in terms of post-colonial movement. In the case of this paper, however, focus is on commercial translation as the students involved in this research are business English majors.

Q4.2. I advise carefully editing the article for English language writing and fluency.

A4.2: The article was sent to a native English-speaking American editor for further revision. Hopefully the writing and fluency have been improved.

Q4.3. Also be sure to define transcreation in the first page or two of the article; this word is very unusual in English.

A4.3: The term transcreation has now been defined at the beginning of the article.

Q4.4. Please check your calculations and use of validity factor analysis and Cronbach’s alpha with a statistician to confirm it is appropriate and correct

A4.4: A statistician has been consulted and the statistics are confirmed to be appropriate and correct. 

Q4.5. There should be more explanation about what statistical tests were used in figures 22 and 23 and you think they are the appropriate tests and what the reasons were to conduct them.

A4.5: Upon the advice of the statistician, a key step in operating the software in connection with the calculations is added in the paper, as shown in Fig 19. The results of the reliability test and validity test involved in those two figures (now Figs 21 and 22 respectively in the revised manuscript) are explained in detail. The two tests are appropriate here and they were conducted for good reasons, as given at the beginning of the section in the revised manuscript: According to D.A. de Vaus (Surveys in Social Research. 5th ed. Crows Nest, NSW: Allen & Unwin; 2002.), three most important principles should be followed when designing survey questions: reliability, validity and question discrimination (currently called discriminability) (p.96). Therefore，three statistical tests for reliability, validity and discriminability were conducted to ensure that the survey results are trustworthy.

Part 2: Response to the First Reviewer Dr. Naji Abi-Hashem

I’d like to thank Dr. Naji Abi-Hashem for your detailed comments and suggestions as well as your kindness and patience in our several email communications. 

Q1. The Abstract and Introduction sections are not easy to read. Opening paragraph is too technical for the general readers. 

A1: The suggested version has been adopted with thanks.

Q2. Not flowing structure and content. Authors dive into the subjects w/o adequate definition of terms or gradual building the ground stage. For example, I had to look up the word transcreation online to exactly understand what the authors mean by it and what are they talking about!?! A brief definition of terminologies is essential to any scholarly document before engaging in any detailed discussion. 

A2: The reviewer's suggestion has been accepted. And Google's definition recommended by the reviewer and author's own definition are presented at the beginning of the Introduction and issues concerned are discussed to gradually build the ground stage.

Q3. The Conclusion has more flow and clear insight than the opening & the introduction. I suggest the authors include some of that clear description in this paper early on so the educated reader can have an idea of the purpose and direction of this manuscript.

A3: Part of that clear description in the Conclusion is now presented in the Introduction. 

See Lines 260-271 in the revised manuscript. 

Q4. Also seems to me that the author’s mother language is not American-English, thus the discrepancies in the text and lack of flow and connectivity. So, the manuscript can benefit from revision to be made by a couple of English-speaking educators who are also familiar with the subject matter.

A4: The article was sent to a native English speaking American editor for revision. Hopefully the flow and connectivity have been improved.

Q5. Too many comas in the text ,,,,,,,, making sentences bit fragmented

A5: The number of comas has been greatly reduced for better coherence.

Q6. The term Student Translator perhaps should be hyphenated for better reading and conceptualization of this specific target population: student-translator(s) or trainer-translators

A6: The term "student translators" without a hyphen are more frequently used by scholars of translation studies. As for the second term, the common form is translator trainee or translation trainee without a hyphen either, referring to translators or students who receive translation training or education, as indicated in the Google.

Q7. Text will better read if authors start some paragraphs with “According to so & so (year),

A7: Revisions have been made as suggested.

See Lines 134 ,349 etc. in the revised manuscript.

Q8. The statement of the problem is not clear! Because there are online Apps or engines that engage in translation from local-national into global or international languages, does not necessarily eliminate the need for the human factors (translators) like any other discipline or market item, available in person in tangible forms and also electronically in online forms.

A8: As shown in the revised manuscript, a common mode of human–machine interaction during the MT process is like this: First input the source text; then the MT produces an output as a draft that is corrected by a human translator. This process is often referred to as post-editing (MTPE, or simply PE). Therefore, the need for human factor is not at all eliminated. But the sense of self-worth felt by human translators is greatly reduced. Now that AI takes over a big share of repetitive or routine work, human translators need to focus instead on creativity. 

The problem with professional and student translators is now clear. While working in parallel or rather in collaboration with AI or machine translation, many of them feel devalued and do not know how to react. 

As illustrated by the literature review, transcreation could be the best solution to the problem.

See Lines 176-206 in the revised manuscript.

Q9. Maybe this very paper has been translated into English with the help of computer software program or similar engine.

A9: I am very sorry for the language of the manuscript. As this author's mother tongue is not English, it is understandable that this article gives the impression that it was translated with the help of a computer software.

Q10.The paper quickly moves to become a Case Study - not a discussion/analysis of a major theme topic. Then moves to talk about EDITING on top of the primary subject Translating.

A10: As shown in the section “Research design", a discussion is now given of the major theme topic that transitions naturally to the selection of one-shot case study.

As mentioned in A8 to Q8, a common mode of human–machine interaction involves a process often called post-editing. Hence, editing is discussed in this sense, which is part of the translating process.

Q11.The research method and survey, followed by statistics, then interpretations, all seem reasonable sequences and important steps.

A11: As suggested, those steps are followed in the revised manuscript.

Q12. Then the authors bring up the cross-cultural issues and dynamics, which are extremely important and organic part of any interaction, relationship, and communication: verbal or non-verbal, overt or covert, explicit or implicit. Here the paper will benefit from citing some major definitions of CULTURE, citing some insights/works in the relevant fields.

A12: Some insights and definitions of culture are cited as advised.

See Lines 95-148 in the revised manuscript.

Q13. Also some literature review needed re: human-machine interaction, experts skills & productivity versus smart machines & computers abilities/productivity. There are a lot of research and insights there.

A13: Some literature review has been done in some of those areas.

See Lines 162-202 in the revised manuscript.

Q14. In the discussion section, it is not clear why the authors focused only on some universities in UK to check whether their MA program incorporates Transcreation, to find only 4 of them do. Since the DGT is the credentialing agency for the EMT in Europe, why not compare that in the whole continent.

A14: As mentioned in the revised manuscript, it would be very interesting to compare EMT programs at all the universities across the continent in terms of transcreation, which, however, is not the focus of this paper. This part is just to illustrate whether transcreation is covered at institutions of higher education. Therefore, only UK study programs are taken for example.

Q15.It is also common sense that we as trained human students or accomplished experts/scholars alike are aware of the new competition and subtle abilities of computers-soft wares and apps – so we adjust, learn to maneuver, and work in parallel rather one eliminates the other. SO, what is novel about this study and its finding or recommendation? Not clear!

A15: Surely, humans and AI-driven machine translation will work in parallel or rather in collaboration with each other. But humans tend to feel devalued in the age of AI. As mentioned in the Conclusion in the revised manuscript, this paper contributes in two ways: theoretically, it proposes a new definition of transcreation although no further elaborations have been made; on the practical side, it appeals for a shift of the focus in translator training to transcreation and incorporating transcreation into the translation syllabus and translator training so that the employability of student translators can be boosted on the one hand and sustainable development in translation courses and programs can be maintained in institutions of higher education on the other. Therefore, theoretical and especially pedagogical contributions are made.

Q16.Text needs revision for better language, structure, flow, and cleaning of punctuations.

A16: As mentioned in A4 to Q4, this article was sent to a native English-speaking American editor for further revision. Hopefully the revised manuscript has improved in those respects.

Q17.The tables and inserts within the text are way too many, so I am not sure how professional to keep so many images interrupting the text. Otherwise they need to be reorganized in better way.

A17: To keep the images or figures from interrupting the text, most of them are put at the end of the article.

Q18.The Reference section /Bibliography is too short for such a compound subject. Definitely, it is not in APA style yet looks bit crowded.

A18：Thanks to the help of Dr. Naji Abi-Hashem, this author has been able to access a lot of important references and insights. The number of listed references has thus more than doubled, totaling 69.

PLOS uses the reference style outlined by the International Committee of Medical Journal Editors (ICMJE), also referred to as the “Vancouver” style, as specified in the webpage of the journal:https://journals.plos.org/plosone/s/submission-guidelines#loc-references.

Part 3: Response to the Second Reviewer Dr. Sayyed Mohsen Fatemi

I’d like to thank Dr. Sayyed Mohsen Fatemi for your positive comments and suggestions as well as your kindness and patience in our email communications.

Q1: The article needs minor revision. It needs to focus on cultural and political discourse of education and postcolonial movement. The article lacks theoretical grounds that focus on the discursive parlance of education and its implications for marginal versus core meanings.

A1: A Google search for "postcolonial movement" has produced millions of results. Of those, two definitions are interesting.

One: Broadly a study of the effects of colonialism on cultures and societies. It is concerned with both how European nations conquered and controlled "Third World" cultures and how these groups have since responded to and resisted those encroachments (https://www3.dbu.edu/mitchell/postcold.htm).

Two: Postcolonialism (or Postimperialism) is the critical academic study of the cultural, political and economic legacy of colonialism and imperialism, focusing on the impact of human control and exploitation of colonized people and their lands. More specifically, it is a critical theory analysis of the history, culture, literature, and discourse of (usually European) imperial power (https://en.wikipedia.org/wiki/Postcolonialism).

In the postcolonial movement, Edward W. Said(1979,p.108) was concerned about “equality” in general while Paulo Freire(2000,p.139) paid particular attention to “equality of all individuals” in education. John Willinsky (1998) addresses various issues like identity and racial inequality(p.5,p.163) in education.

The focus of this paper, however, is not about education in general nor is it about education in the post-colonial context in particular. Perhaps the word "education" in the title of this paper is misleading or I might have misunderstood postcolonialism. The major concern of this paper is translator training and its sustainability.

I am very grateful to Dr. Sayyed Mohsen Fatemi for pointing out the lack of theoretical grounds in this article. Since translation is a cross-cultural or inter-cultural activity, cross-cultural psychology is used in the revised manuscript as a frame of reference against which many terminologies were used and relevant analyses were conducted in this paper.

References:

Freire, P. (2000). Pedagogy of the Oppressed, 30th Anniversary Edition. Continuum. 

Said, E. W. (1979). Orientalism (1st Vintage Books ed). Vintage. 

Willinsky, J. (1998). Learning to Divide the World: Education at Empire’s End. U of Minnesota Press.

---

## [Editor Report · Decision Letter 1]

13 Mar 2023

Sustainability of Translator Training in Higher Education

PONE-D-22-30289R1

Dear Dr. Zhu,

We’re pleased to inform you that your manuscript has been judged scientifically suitable for publication and will be formally accepted for publication once it meets all outstanding technical requirements.

Kind regards,

Grant Rich, Ph.D.

Academic Editor

PLOS ONE
---

## [Editor Report · Acceptance letter]

10 Apr 2023

PONE-D-22-30289R1 

Sustainability of translator training in higher education 

Dear Dr. Zhu:

I'm pleased to inform you that your manuscript has been deemed suitable for publication in PLOS ONE. Congratulations! Your manuscript is now with our production department. 

Kind regards, 

on behalf of

Dr. Grant Rich 

Academic Editor

PLOS ONE